# Differences in the Factor Structure of the Eating Attitude Test-26 (EAT-26) among Clinical vs. Non-Clinical Adolescent Israeli Females

**DOI:** 10.3390/nu15194168

**Published:** 2023-09-27

**Authors:** Zohar Spivak-Lavi, Yael Latzer, Daniel Stein, Ora Peleg, Orna Tzischinsky

**Affiliations:** 1Faculty of Social Work, The Max Stern Yezreel Valley College, D.N. Emek Yezreel 1930600, Israel; 2Faculty of Social Welfare and Health Sciences, University of Haifa, Haifa 3498838, Israel; ylatzer@univ.haifa.ac.il; 3Eating Disorders Institution, Psychiatric Division, Rambam Health Care Campus, Haifa 31096, Israel; 4Sackler School of Medicine, Tel Aviv University, Tel Aviv 69978, Israel; prof.daniel.stein@gmail.com; 5Safra Children’s Hospital, Sheba Medical Center, Ramat Gan 52621, Israel; 6Education and School Counseling Departments, Max Stern Yezreel Valley College, Yezreel Valley 1930600, Israel; 7Department of Behavioral Sciences, The Max Stern Academic College of Emek Yezreel, Emek Yezreel 1930000, Israel; orna@yvc.ac.il

**Keywords:** EAT-26, assessment, eating disorders, clinical and non-clinical populations

## Abstract

In recent years, the diagnostic definitions of eating disorders (EDs) have undergone dramatic changes. The Eating Attitudes Test-26 (EAT-26), which is considered an accepted instrument for community ED studies, has shown in its factorial structure to be inconsistent in different cultures and populations. The aim of the present study was to compare the factor structure of the EAT-26 among clinical and non-clinical populations. The clinical group included 207 female adolescents who were hospitalized with an ED (mean age 16.1). The non-clinical group included 155 female adolescents (mean age 16.1). Both groups completed the EAT-26. A series of factorial invariance models was conducted on the EAT-26. The results indicate that significant differences were found between the two groups regarding the original EAT-26 dimensions: dieting, bulimia and food preoccupation, and oral control. Additionally, the factorial structure of the EAT-26 was found to be significantly different in both groups compared to the original version. In the clinical group, the factorial structure of the EAT-26 consisted of four factors, whereas in the non-clinical sample, five factors were identified. Additionally, a 19-item version of the EAT-26 was found to be considerably more stable and well suited to capture ED symptoms in both groups, and a cutoff point of 22 (not 20) better differentiated clinical samples from non-clinical samples. The proposed shortening of the EAT from 40 to 26 and now to 19 items should be examined in future studies. That said, the shortened scale seems more suited for use among both clinical and non-clinical populations. These results reflect changes that have taken place in ED psychopathology over recent decades.

## 1. Introduction

### 1.1. Eating Disorders—Past and Present

Eating disorders (EDs) are severe psychiatric illnesses [1,2], and their rate has steadily increased over the past four decades both in Western and non-Western countries [3,4]. The lifetime prevalence of EDs (according to the DSM 5 (Diagnostic and Statistical Manual of Mental Disorders) diagnostic criteria [5]) is estimated to be about 1–1.5% for anorexia nervosa (AN); 1–2% for bulimia nervosa (BN); and 1–3.5% for binge eating disorder (BED) [6,7]. The prevalence of pathological eating behaviors and a low self-esteem and negative body image among female adolescents and young women, in different studies worldwide, has been estimated to be about 40–70% [8].

Over the past five decades, the clinical picture of EDs, as reflected in the different DSM classifications, has undergone dramatic changes [9]. Bulimia nervosa was added in the early 1980s; BED was added as a provisional diagnosis in the early 1990s; and normal-weight purging disorder, night eating syndrome, and avoidant restrictive feeding and eating disorder (ARFID) for young children were only added around 10 years ago [10].

The changes in the DSM criteria reflect the changes in the prevalence of the different clinical presentations of EDs over the past several decades, and are related, in part, to several important socio-cultural changes. These include the plentitude of food in Westernized countries, increased globalization processes, the exposure of people in non-Westernized countries to societal pressures, and thus, an increase in the risk for EDs [11,12], and changes involved in the distribution and influence of mass media with respect to body image and dieting behaviors, which have increased dramatically in recent decades [13].

In light of these changes, the early detection of individuals who are at a high risk of developing EDs is critical for improvements in prevention, treatment, and prognosis, and the reduction in chronicity [14]. Indeed, over many years, attempts have been made to develop improved assessment tools to better identify these different types of at-risk ED groups [15]. 

### 1.2. Self-Reported Screening Tools for Disordered Eating Behaviors

Over the past four decades, a series of assessment tools has been created for the purpose of identifying the presence of at-risk groups for ED symptoms and ED severity. These include the Eating Disorder Inventory (64–91 items depending on the version [16,17]); the Eating Disorder Examination, Screening Version (8 items [18]); the Eating Disorder Examination Questionnaire (36–41 items [19]); the SCOFF Questionnaire (5 items [20]), and the Eating Attitudes Test (EAT; 26–40 items [21,22,23]). 

The EAT questionnaire is commonly used by clinicians and for research purposes in the field of Eds. Its original version, the EAT-40 [22], was developed when AN was characterized mainly by restrictive behaviors [24]. It included seven factors [25]: food preoccupation, a drive for thinness and body-image-related preoccupations, vomiting and laxative abuse, dieting, slow eating, covert eating, and perceived pressure to gain weight. The answers were rated on a 6-point Likert scale, with a cutoff point of 30. A score higher than 30 was considered to identify disturbed eating behavior.

The EAT-40 was found to be valid in patients with AN in a community sample [21]. Nevertheless, it yielded high percentages of false positive scores among potentially high-risk groups—for example, 29% among ballet students and 27% among modeling students [22]. Notwithstanding these limitations, the EAT-40 was considered an effective screening questionnaire for identifying groups at risk of developing EDs [15].

Over the years, the questionnaire was shortened to a 26-item version [21]. The short version includes three scales: dieting, bulimia and food preoccupation, and oral control (i.e., showing self-control overeating, including in conditions when there are environmental pressures and perceived pressures to eat and gain weight). The answers are rated on a 6-point Likert scale, with a cutoff point of 20 or higher showing disturbed eating behavior. Both cutoff points (20 for the EAT-26 and 30 for the EAT-40) have been supported by studies conducted among clinical and non-clinical samples and may assist in identifying people who are at risk of developing an ED [26,27,28,29,30]. The two versions of the EAT have been found to be comparable in the identification of disturbed eating in the general population [15]. 

The EAT-26 was translated, validated, and adapted into many languages, including Arabic [31], Japanese [32], Italian [33], and Hebrew [34]. In addition, it was examined among diverse ethnic groups, including in Israel [15,25,35], and was declared as the screening instrument of choice for the identification of disordered eating in the general population by the National Eating Disorders Screening Program and by the National Mental Illness Screening Project in 1999 [36]. 

There are, however, several reasons to suggest that a cutoff point of 20 is no longer a valid cutoff point. 

Notwithstanding the generally accepted reliability of the EAT-26 in studies about disordered eating in community populations, concerns have arisen in recent years among clinicians and researchers regarding the use of this instrument. The main suggested reason for this concern stems from recent results of a study conducted by our group in Israel [35], emphasizing that the EAT-26 is used differently, and its results are interpreted differently, in tradition-oriented sub-populations. The results show different factors in different ethnic groups, most of which did not correspond with the original EAT-26 three-factor structure. The analysis yielded two main factors among Israeli Jews, four main factors among Israeli Muslim Arabs, and three main factors among Israeli Christian Arabs, revealing the inconsistencies that were found in its factor structure [37]. 

Similarly, a few studies in English-speaking countries have reported either three, four, or five factors, with the number of items ranging from 16 to 25 [15]. In non-English-speaking samples, four to six factors were observed, and a new factor was also identified: an awareness of food preoccupation [38]. As for the Hebrew version, it yielded the three original factors, but a fourth factor, an awareness of food preoccupation, was also identified [28]. 

Another suggested reason is related to the finding that both the EAT-40 and the EAT-26 mainly include items assessing restricting preoccupations and behaviors, with only a few items referring to binging/purging behaviors. This situation likely arose because, at the time of the construction of the two scales, in the late 1970s and early 1980s, respectively, the most studied ED was AN of the restricting type. Following the changes observed over the past decades in the symptoms and psychopathological characteristics of EDs in general and AN in particular, it appears that it is mainly the total score of the EAT-26, rather than its factors, that can be used for the assessment of the severity of ED symptoms in patients with AN, in addition to changes in symptom severity following treatment. 

To the best of our knowledge, there have been no studies in which the factor structure of the EAT-26 has been compared between clinical and non-clinical populations.

Hence, several questions have arisen: 1. Does the EAT-26 questionnaire still clearly distinguish between a healthy population and a population with EDs in all its factors? 2. Can the EAT-26 be used today, in 2023, as a screening tool to identify risk groups for the development of EDs in different ethnic populations? 3. Does the cutoff point of 20 allow for a distinction to be made between clinical and non-clinical populations in 2023? 4. Is the factor structure of the EAT-26 similar in clinical and non-clinical populations? 

Based on these questions, the main goal of the present study was to examine the factor structure of the EAT-26 questionnaire and the relevant cutoff score in two populations of adolescent girls in Israel: a clinical population and a non-clinical population.

We hypothesized the following:
A difference would be found between a clinical population and a non-clinical population in the factor structure of the EAT-26.The clinical group would show a factor structure that is more like the original EAT-26 (three factors) than the non-clinical group would. In accordance with the first hypothesis, a difference would be found between the current EAT-26 cutoff point, reflecting the presence of pathological eating preoccupations and behaviors, and the original cutoff point (20).

## 2. Materials and Methods

### 2.1. Participants and Procedure

Two groups were included in the study: a clinical group and a non-clinical group. The clinical group consisted of 207 Jewish Israeli female adolescents who were hospitalized in the Specialized Eating Disorders (EDs) Adolescent Inpatient Department at Safra Children’s Hospital, Sheba Medical Center, Tel Hashomer, Israel for the treatment of EDs between the years 2008 and 2020 (masked for review). Of these, 206 participants were under the age of 18, and one participant was 19 years old. She was included in the study because her hospitalization began before she turned 18 and continued until she was 19 years old. It should be noted that, in Israel, the psychiatric care of children is considered to extend up to the age of 21. 

The mean age of the patients was 16.1; SD (1.3); and range (12.3–19.0). The most prevalent diagnosis was AN restricting type (n = 94; 45.4%); followed by AN binge/purge type (n = 45; 21.7%); BN (n = 33; 15.95%); and atypical EDs (22; 10.6%). The most prevalent comorbid psychiatric diagnosis was depression (39; 18.8%); followed by ADHD (25; 12.1%); anxiety (10; 4.8%); and obsessive–compulsive disorder OCD (9; 4.3%) (see Table 1).

The EAT-26 questionnaires of the clinical sample were filled out during the course of various studies that were conducted in this department, and were all approved by the IRB of the Sheba Medical Center (#2755). The need for informed consent for this specific study was waived due to the study’s retrospective nature—namely, a review of electronic medical records. The only other details presented in this study from the medical records were the participants’ ages, weights, and heights. The EAT-26 questionnaires were filled out within the first two weeks of hospitalization—that is, when the adolescent was in an acute ED state. 

In this department, the diagnosis of an ED is determined by experienced child and adolescent psychiatrists using semi-structured interviews, based on the DSM-IV and DSM-5 criteria. Patients were excluded from study participation if their ED diagnosis was not confirmed unanimously in clinical team meetings of the inpatient department. All patients diagnosed with an ED via the DSM-IV were also re-diagnosed with the same ED when their charts were re-evaluated according to the DSM-5. Comorbid psychiatric diagnoses were similarly determined, using semi-structured interviews, based on the DSM-IV and DSM-5 criteria. 

The second group (i.e., the control group) consisted of 156 Jewish Israeli non-clinical female participants of a similar age range to that of the clinical group. One participant was 21 years old, and therefore, was omitted from the sample. Between the years 2011 and 2019, four high schools were recruited for the purpose of the study after receiving ethical approval from the institutional review board (IRB) of the first author’s college, the Ministry of Education, the high school principals, the teachers, and the adolescents and their parents. In each high school, one class was sampled from each age group, and those who agreed to participate filled out the questionnaires at school. Informed consent was obtained from the adolescents and their parents. The mean age of the control group ranged from 12 to 18 years. The girls in the clinical group had statistically significantly lower BMIs than their healthy counterparts (*p* < 0.001). 

In both study groups, completion of the questionnaires was voluntary, and respondents were told they could stop participating at any point. All participants were assured of anonymity and confidentiality. Controls were excluded if they reported an ED via an open yes/no question. They were also asked to report their weight and height. Only those controls fulfilling these two criteria were included in the study. 

### 2.2. The EAT-26 Instrument

The Eating Attitude Test (EAT-26) [21] is a screening instrument that is commonly used to measure eating attitudes. It comprises 26 items, and scoring is completed on a 6-point scale. The six available response options are “always”, “usually”, “often”, “sometimes”, “rarely”, or “never”. The subscales are dieting, bulimia and food preoccupation, and oral control. Scores range between 0 and 78 points, and higher scores indicate greater pathology. A score of 20 or higher indicates a clinically significant eating pathology level; it is referred to as an EAT-26 “positive score”. The EAT-26 has demonstrated high reliability and consistency, and the initial Cronbach’s alpha was 0.90 for the total EAT-26 [21]. In the current study, total internal consistency: α = 0.86; dieting factor: α = 80; bulimia and food preoccupation factor: α = 67; and oral control factor: α = 56.

### 2.3. Data Analysis

To estimate whether the Israeli clinical and non-clinical samples manifest EDs in the same way, we conducted a series of factorial invariance models on the EAT-26. In the first model, we tested for *configural invariance* (i.e., pattern invariance), in which one assesses whether similar items measure each construct across groups (i.e., whether EAT-26 dieting, bulimia and food preoccupation, and oral control clusters consist of the same items in clinical and non-clinical samples). In configural invariance models, items are loaded on predefined latent factors, as in Confirmatory Factor Analysis (CFA), such that items’ loadings and intercepts are freely estimated for each group (i.e., only the factorial structure is fixed, while the loadings and/or intercepts could be different for each group). Good model fit would support configural invariance. Model fit was estimated via Comparative Fit Index (CFI), Tucker–Lewis Index (TLI), Root Mean Square Error of Approximation (RMSEA), and Standardized Root Mean Square Residual (SRMR). CFI and TLI > 0.90 and RMSEA and SRMR < 0.07 are acceptable. Next, we tested for *metric invariance* (i.e., weak invariance), in which one assesses whether the constructs’ factor loadings are similar across groups (i.e., loadings are constrained to be equal across groups); attaining invariance of factor loadings suggests that the constructs have the same meaning to participants across groups. Metric invariance is assessed by comparing the fit of the configural model with that of the metric invariance model; a non-significant chi-squared test would support metric invariance. Of note, metric invariance is not enough to justify the comparison of group means. Next, we tested for *scalar invariance* (i.e., strong invariance), in which one assesses whether items have the same intercepts (i.e., both loadings and intercepts are constrained to be equal across groups). Non-invariance of intercepts may be indicative of potential measurement bias and suggests that there are larger forces such as cultural norms or developmental differences that are influencing the way participants are responding to items across groups. Attainment of scalar invariance justifies comparison of group means. Scalar invariance is assessed by comparing the fit of the metric model with that of the scalar invariance model; a non-significant chi-squared test would support scalar invariance. All models were estimated with the *lavaan* Structural Equation Modeling R package. Missing data were handled with the Full-Information Maximum Likelihood (FIML) method.

Following the assessment of factorial invariance (or lack thereof) between the clinical and non-clinical samples, we employed Exploratory Graph Analysis (EGA; [39]) using *EGAnet* R package—a network psychometrics method that uses undirected network models for the assessment of psychometric properties of questionnaires. Exploratory Graph Analysis was used to verify the number of factors using graphical lasso [40] and the items that are associated with each factor. Network loadings, which are roughly equivalent to factor loadings, are reported using *net.loads()*, with suggested general effect size guidelines for network loadings of 0.15 for small effect sizes, 0.25 for moderate effect sizes, and 0.35 for large effect sizes [41]. The number of factors corroborated other traditional methods—parallel analysis (PA), Velicer’s minimum average partial (MAP) test, and the comparison data approach [42]. Next, to examine the stability of the EGAs, we followed the analysis using Bootstrap EGA with 5000 resampling cycles. Finally, we used the novel Unique Variable Analysis (UVA [41]) for detecting redundant items in the EAT-26, and used the *item Stability()* function to detect highly unstable items. 

In the final part of the results, we examined the effectiveness of the revised EAT questionnaire in differentiating between clinical and non-clinical samples. To achieve this, we calculated the optimal clinical cutoff point by bootstrapping the optimal cutoff point while maximizing the sensitivity and specificity (i.e., highest Youden’s index: sensitivity + specificity—1). We also reported the suggested indexes of the “number needed to diagnose” (NND [43]) (i.e., the number of patients who need to be examined in order to correctly detect one person with the disease of interest in a study population of persons with and without the known disease); “number needed to misdiagnose” (NNM; [1]) (i.e., the number of patients who need to be tested in order for one to be misdiagnosed by the test); and the “likelihood to be diagnosed or misdiagnosed” (LDM [44]), with higher values of LDM (>1) suggesting that a test is more likely to diagnose than misdiagnose. 

## 3. Results

### 3.1. Assessing Configural Invariance

To assess whether similar items measure the EAT-26 constructs across groups (clinical and non-clinical), we conducted a configural invariance model. The model had a poor fit with the observed data, including *χ*^2^_(227)_ = 1041.95, *p* < 0.0001, *CFI* = 0.77, *TLI* = 0.74, *RMSEA* = 0.13 (90% confidence intervals (CIs) of 0.12 and 0.14), and *SRMR* = 0.10, indicating that the EAT-26 control clusters of dieting, bulimia and food preoccupation, and oral control do not consist of the same items across the groups. We followed the modeling with *modindices()* to examine whether the covariates between the items might improve the model’s fit. Eight covariates were identified, yet remodeling the suggested factorial construct with these covariates did not improve the fit, as seen in the following: *χ*^2^_(288)_ = 1094.54, *p* < 0.0001, *CFI* = 0.77, *TLI* = 0.74, *RMSEA* = 0.12 (90% CIs of 0.11 and 0.12), and *SRMR* = 0.10. In other words, the factorial structure of the EAT-26 was found to be significantly different from the original EAT-26 version in Israeli clinical and non-clinical groups (i.e., the configural invariance did not hold). As a result of this finding, we did not further examine the presence of matric or scalar invariances. 

### 3.2. Exploratory Graph Analysis

To assess the factorial structure of the EAT-26 within each group, we conducted an EGA separately for the clinical and non-clinical samples. The EGA network results are presented in Figure 1, and the network loadings are shown in Table 2. 

### 3.3. Clinical Sample

The analyses indicated that the factorial structure of the EAT-26 in the clinical group consisted of four factors named “Weight preoccupation”, consisting of eight items; “Binge/purge behaviors and concerns of others”, consisting of four items; “Dieting and restricting symptoms”, consisting of six items; and “Eating-related concerns”, consisting of eight items (with item 26, “I enjoy trying new rich foods”, and item 19, “I display self-control around food”, loading only weakly). A four-factor solution was corroborated by two additional analyses—parallel analysis (eigen values of 7.23, 3.97, 2.04, and 1.45 for the four factors) and Velicer’s MAP (squared: 0.016; fourth power: 0.0009). Conversely, the comparison data estimation suggested that five factors must be retained (as compared with one to seven factors). 

When estimating the stability of the EGA by bootstrapping with 5000 resampling cycles, the analysis indicated high stability (SE = 0.67), with the CI for the number of factors ranging from 2.69 to 5.31. In addition, the four-factor solution was prevalent in 63.90% of the bootstrap samples, with 23.48% producing a five-factor solution (and 8.64% producing a three-factor solution, and 3.66% producing a six-factor solution). A confirmatory factor analysis (CFA) that was used to corroborate the EGA solution verified the factorial structure as follows: *χ*^2^_(82.99)_ = 173.96, *p* < 0.01, *CFI* = 0.96, *TLI* = 0.95, *RMSEA* = 0.073 (90% confidence intervals (CIs) of 0.065 and 0.081), *SRMR* = 0.08. The CFA is presented in Figure 2a. 

### 3.4. Non-Clinical Sample

The analyses indicated that the factorial structure of the EAT-26 in the non-clinical group consisted of five factors: “Weight concerns”, consisting of four items; “Eating-related concerns”, consisting of five items; “Food controls one’s life”, consisting of four items; “One’s own and others’ control over the person’s eating”, consisting of five items; and “Dieting”, consisting of five items, with item 25 (“have the impulse to vomit after meals”), item 7 (“particularly avoid foods with a high carbohydrate content”), and item 19 (“display self-control around food”) not loading significantly on any of the factors. A five-factor solution was corroborated by three additional analyses—parallel analysis, Velicer’s MAP (squared: 0.00178), and comparison data (as compared with one to seven factors). 

When estimating the stability of the EGA by bootstrapping with 5000 resampling cycles, however, the analysis indicated instability (SE = 0.79), with the CI for the number of factors ranging from 3.44 to 6.56. Although the five-factor solution was the most prevalent (42.94% of the bootstrap samples), a four-factor solution was also frequent, with 40.36% of the sample. This instability might stem from two main processes: redundant items and items with high instability. A unique variable analysis revealed several redundant items, namely, in the presence of item 3 (“find (UVA) myself preoccupied with food”), item 21 (“give too much time and thought to food”) is redundant. In the presence of item 4 (“have gone on eating binges where I felt that I may not be able to stop”), item 18 (“feel that food controls my life”) is redundant. Finally, in the presence of item 8 (“feel that others would prefer if I ate more”), item 20 (“feel that others pressure me to eat”) is redundant.

We omitted the redundant items, conducted an additional EGA with bootstrapping, and used the *itemStability()* function (see Figure 3). 

The analysis revealed that items 3, 4, 19, and 25 were all unstable. We removed the unstable items and repeated the quality testing (i.e., bootstrapping and testing item stability), and found no additional problems. The final 19-item version of the EAT was found to be considerably more stable than the original 26-item version (SE = 0.55; CI 2.91, 5.09), with 73.40% of the bootstrap samples producing a four-factor solution. A confirmatory factor analysis (CFA) that was used to corroborate the EGA solution verified the factorial structure as follows: *χ*^2^_(42.70)_ = 82.72, *p* < 0.01, *CFI* = 0.96, *TLI* = 0.95, *RMSEA* = 0.063 (90% confidence intervals (CIs) of 0.052 and 0.072), and *SRMR* = 0.08. The CFA of the suggested EAT-19 questionnaire for the non-clinical samples is presented in Figure 2b, and the network loadings are shown in Table 3.

To examine the use of the EAT-19 in the clinical sample as well, we appraised its structure and stability in this population. The EGA produced a four-factor solution (see Figure 4 and Table 4), showing adequate stability when administering a bootstrap EGA (*SE* = 0.73; CI 2.58, 5.42), with 54.02% of the samples reproducing the solution. In addition, 18 out of the 19 items had adequate stability, with item 9 showing only a 51% replication rate. Overall, the EAT-19 seems to be well suited to capture ED symptoms among both non-clinical and clinical samples alike. 

### 3.5. Effectiveness of Using the EAT-19 as a Diagnostic Test

Bootstrapping the optimal cutoff point of the suggested EAT-19 revealed that using a cutoff point of 21.68 (i.e., practically rounded to 22) produces a maximum Youden’s index of 0.69, with a sensitivity of 83.82% and a specificity of 85.23% (see Figure 5). By using the novel EAT-19 version and the cutoff point of 22, 1.45 patients would need to be examined in order to correctly detect one person with the disease of interest in a study population of persons with and without the known disease (i.e., NND value). In addition, 6.49 patients would need to be tested in order for one person to be misdiagnosed by the test (i.e., NNM value). The overall likelihood to be diagnosed or misdiagnosed is 4.48, which indicates high effectiveness in the diagnosis process. 

## 4. Discussion

This study sought to examine whether the EAT-26, which was developed over 40 years ago as a screening tool for identifying individuals who are at a high risk of developing disordered eating and symptoms of EDs, still meets this purpose in 2023. This question is of great relevance given the many changes in the presentation and distribution of ED symptoms that have occurred since the conception of the EAT-26, as well as recent studies casting doubt on the consistency of its three-factor structure in different cultures. Indeed, the present findings indicate that the original EAT-26 three-factor model is not applicable to young Jewish Israeli women, whether from a community (five-factor model) or from a clinical (four-factor solution) sample. Moreover, we found that a 19-item EAT version shows considerably greater stability than the original EAT 26-item version. Overall, we suggest that the adapted 19-item EAT questionnaire might be suitable for identifying individuals who are at risk of developing disordered eating behaviors in clinical and community populations in Israel. The next step is to replicate our findings in non-clinical and clinical populations in other countries around the globe.

Additionally, our findings support the research hypotheses regarding the differences in the factor structure of the EAT-26 between clinical and non-clinical populations (Hypothesis 1), the greater similarity of the clinical group’s four-factor solution to the original three-factor EAT-26 model vs. the non-clinical group’s five-factor solution (Hypothesis 2), and the difference in the cutoff points for defining pathological eating in both clinical and non-clinical populations in our study (score ≥ 22) vs. the original score of ≥20 (Hypothesis 3; [2]). In this respect, it is of note that the differences in the EAT structure in the present study between the clinical and non-clinical samples are in line with the initial research [22] for the EAT-40 [21] and for the EAT-26, showing, for both scales, significant differences between the clinical samples (AN) and the non-clinical samples. Moreover, in the original studies of the questionnaire, significantly higher percentages of participants in the clinical sample vs. the non-clinical sample scored above the cutoff point of 20.

More specifically, the EGA of the EAT-26 in the clinical group yielded four factors: “weight preoccupation”, consisting of eight items; “binge/purge behaviors and concerns of others”, consisting of four items; “dieting and restricting symptoms”, consisting of six items; and “eating-related concerns”, consisting of eight items (item 26, “I enjoy trying new rich foods”, and item 19, “I display self-control around food”, loaded only to a small extent, and were therefore excluded).

The EGA in the non-clinical sample yielded five factors: “weight concerns”, consisting of four items; “eating-related concerns”, consisting of five items; “food controls one’s life”, consisting of four items; “one’s own and others’ control over the person’s eating”, consisting of five items; and “dieting”, consisting of five items (the loadings of item 25, “have the impulse to vomit after meals”, item 7, “particularly avoid foods with a high carbohydrate content”, and item 19, “display self-control around food” were smaller and therefore excluded).

In recent years, studies in different countries replicated the three-factor structure of the EAT-40 and EAT-26 [45,46]. In a recent study conducted by our group in Israel [35], different factors were observed in different ethnic groups, most of which did not correspond with the original EAT-26 three-factor structure.

The findings of the present study further support this contention in showing differences in the EAT factor structure among clinical populations vs. non-clinical populations.

It should be noted that the previous study, which found that different factors were observed in different ethnic groups in Israel [35], was conducted among adult women, whereas the current study was conducted among teenage girls, and the difference in findings may perhaps be attributed to this age difference. Specifically, it is possible that the questionnaire is experienced differently during adolescence (a period during which there is a high risk for the development of eating disorders) than in adulthood (when there is greater maturity and emotional development).

There are several possible explanations for the differences in the EAT factor structure between the clinical and non-clinical groups. These differences are likely not related to the methodological considerations, as the two groups were of similar age and were studied around the same time period.

One possible explanation for the difference in the EAT-26 between the clinical and non-clinical populations might be the nature of the clinical population. Although there was more than one ED diagnosis among the adolescents in the clinical sample, this group was a more unified group (than the non-clinical group) and was characterized by specific core ED traits. The non-clinical group, for its part, represented other, more diverse non-ED populations, and the female adolescents who made up this group were likely to be more affected by socio-cultural trends (i.e., compared to the female adolescents in the clinical group, who were likely more invested in their illness).

Another possible explanation is that the present results reflect a change in psychopathology in recent decades, whereby many more patients now suffer from the AN binge/purge type rather than the AN restricting type [47] There has also been a significant change in lifestyle habits in recent decades, which is reflected in eating and sports habits, nutrition, and body perception [48]. In addition, there has been a significant change in the media, which broadcasts harmful advertisements related to diets and unreal body image [49]. These advertisements may have harmful effects on the population in general, and on youth in particular, potentially leading to poor body image [50]. These changes may explain various phenomena that have characterized recent years, such as an increased tendency for aesthetic/beauty procedures and surgeries starting at a young age, the use of many harmful techniques for weight regulation, and increases in psychopathology. All of these tendencies may be expressed by both clinical and non-clinical groups [51,52].

### 4.1. Limitations, Directions for Future Research, and Conclusions

#### 4.1.1. Limitations

The present study had several limitations. First, the sample consisted of female adolescents only (between 12 and 19 years), and EDs may manifest differently in young adults, particularly in BN. Going forward, researchers should examine the effect of age on the current findings regarding the EAT-26 in both groups.

Second, the suggested sample size for the CFA is 200, given that the results can be less stable with smaller sample sizes. Although the sample size was adequate for the clinical group, the non-clinical sample size consisted of 155 participants; as such, the results for the non-clinical sample should be seen as preliminary.

Third, the present study was conducted in Israel, which is a “melting pot” of immigrants and a multicultural country that is both very modern and very traditional and is made up of many different cultures, so the results may not represent other Western countries.

#### 4.1.2. Future Research

At this point, it seems that the EAT-26 questionnaire still allows for a distinction to be made between clinical and non-clinical populations of the same culture with the same cutoff points.

We would suggest that the questionnaire be tested in a clinical population compared to a non-clinical population in future studies. We would also suggest that the proposed cutoff point and the 19-question questionnaire be tested in diverse populations in order to validate the findings of the current study more comprehensively and to test the new cutoff point accordingly.

#### 4.1.3. Clinical Implications

The questionnaire probably remains suitable for the identification of disturbed eating in clinical groups, but less so in non-clinical groups, emphasizing the necessity of adapting the tool to account for changes in the presentation of EDs in recent years.

Given that the psychopathology of EDs seems to have changed in recent decades, we propose revising the questionnaires in order to adapt them to the current situation. In our opinion, the current questionnaire indeed provides a suitable response to the psychopathology of EDs in the 21st century.

The current study’s contribution lies in its potential to sharpen clinicians’ ability to identify populations in more difficult clinical situations, to aid them in diagnosis, and to help them identify the severity level, so that the therapeutic approach can be better adjusted. Moreover, the abbreviated questionnaire is easier to fill out than the previous one and is also user-friendly.

## 5. Conclusions

Our findings highlight the differences in the EAT factor structure between our community sample as compared to the original sample, which led to a reduction in the number of scale items from 26 to 19, and led to a change in the cutoff point from 20 to 22.

The EAT, which was developed four decades ago, has thus been reexamined in the current paper and adapted to the requirements of the current era.

The proposed shortening of the EAT from 40 to 26 and now to 19 items must be examined in future studies. However, such a change would render the questionnaire more useful in both clinical and research conditions.

## Figures and Tables

**Figure 1 nutrients-15-04168-f001:**
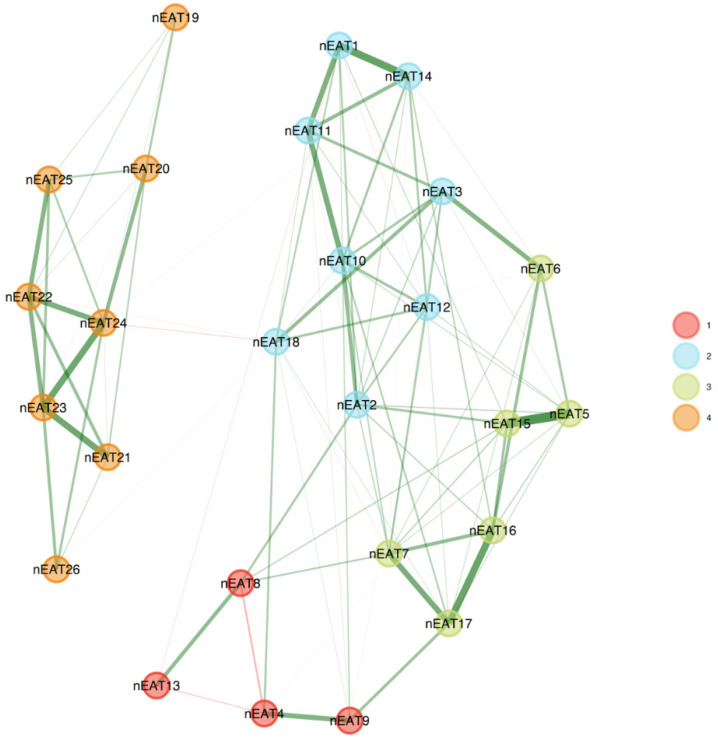
EGA results for clinical (**upper panel**) and non-clinical (**bottom panel**) groups. The factorial structure of the EAT-26 among the clinical group comprised three factors, but with different item configurations than the original EAT-26. The factorial structure among the non-clinical group comprised five factors.

**Figure 2 nutrients-15-04168-f002:**
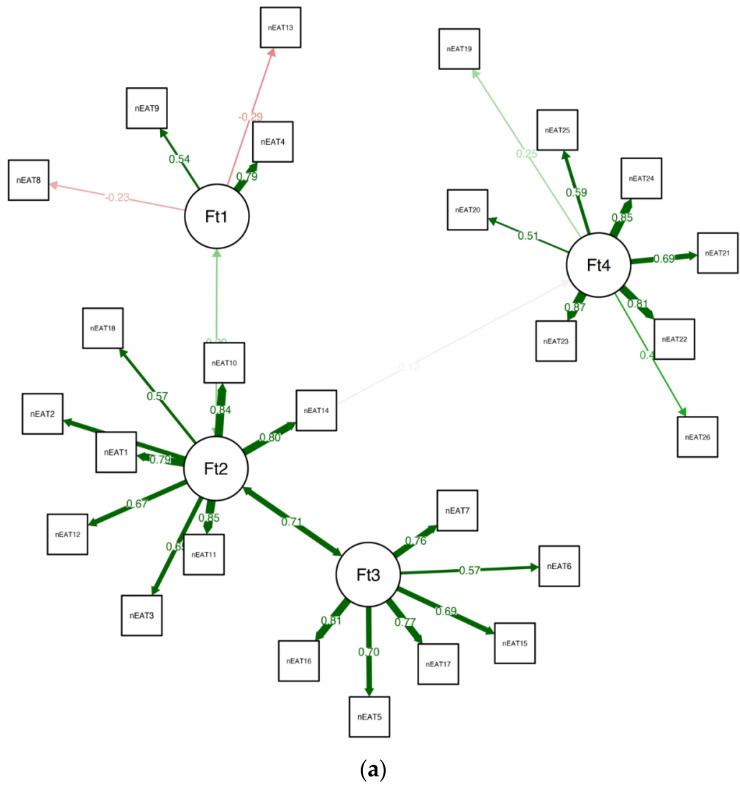
(**a**) The final CFA for clinical samples. (**b**) The final CFA for the suggested 19-item EAT version for non-clinical samples.

**Figure 3 nutrients-15-04168-f003:**
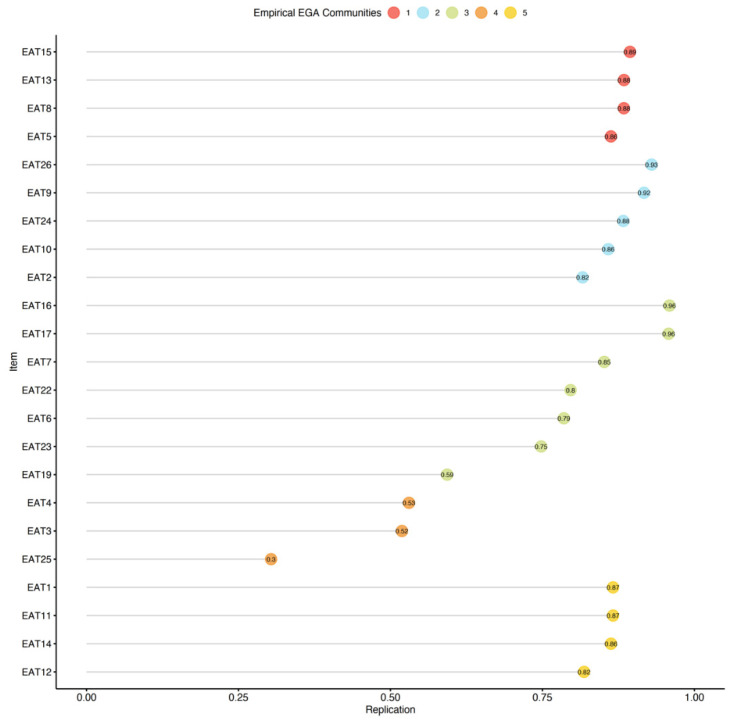
Item stability among the non-clinical sample. Stability below 75% is poor.

**Figure 4 nutrients-15-04168-f004:**
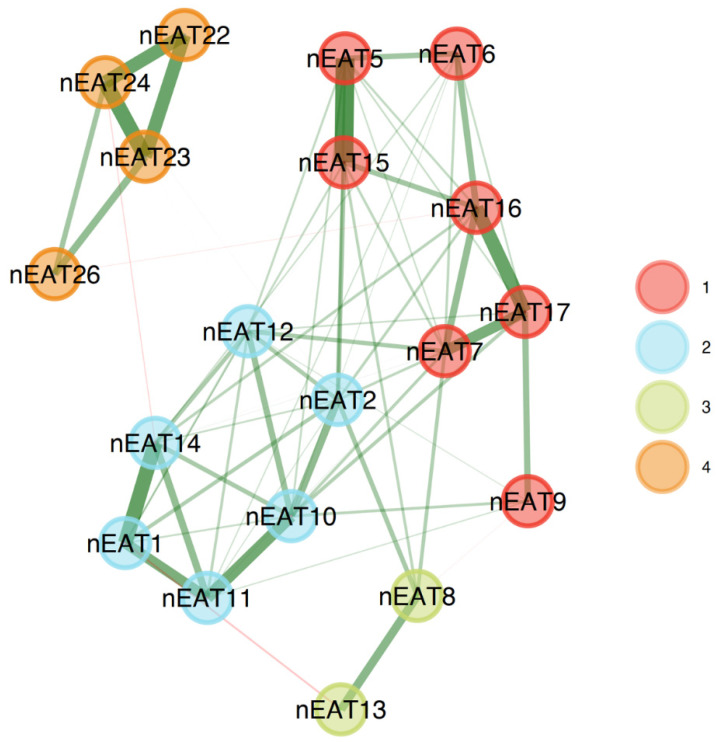
EAT-19 factorial structure among the clinical group.

**Figure 5 nutrients-15-04168-f005:**
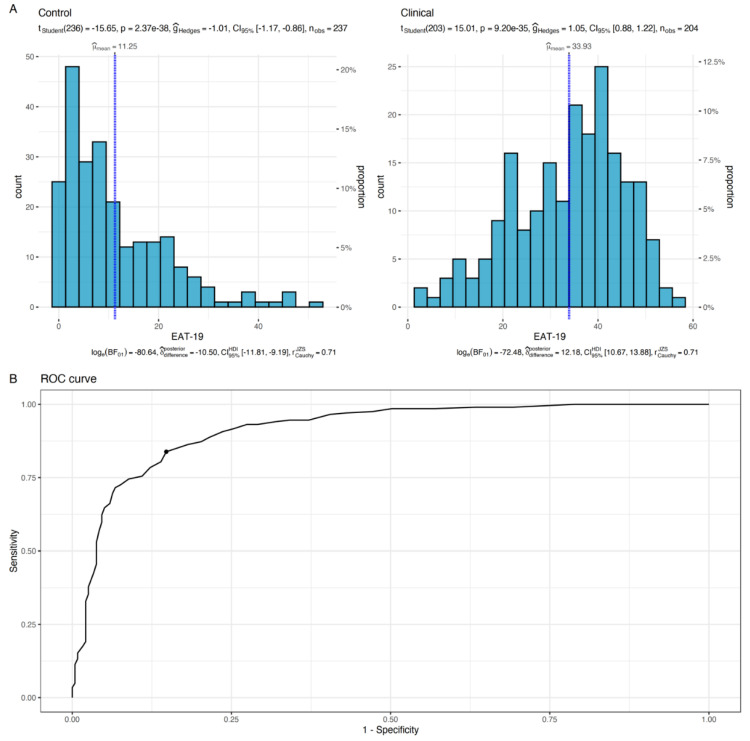
Distribution of EAT-19 scores for the clinical and non-clinical groups (**A**). Vertical blue lines refer to the mean sample score. The test reported in the upper section of panel A examined the difference between the sample mean and the suggested EAT-19 cutoff point (i.e., 22). In (**B**), the ROC curve for the estimation process of the optimal cutoff point is presented, with the black dot indicating the highest Youden’s index.

**Table 1 nutrients-15-04168-t001:** Between-group differences in demographic and clinical parameters.

	Study Group (N = 207)	Control Group (N = 155)	*p*
**Age in years** (range)	16.1 ± 1.3 (12.3–19.0)	16.1 ± 1.6 (12–18)	0.99
**BMI** (range)	17.85 ± 3.46 (10.70–33.60)	21.07 ± 1.61 (14.53–31.64)	<0.001
BMI < 15	35 (16.9)	3 (1.9)	<0.001
BMI > 25	9 (4.4)	31 (19.9)	<0.001
**Diagnosis (n/p)**	94 (45.4)	N/A	
AN	45 (21.7)		
AN—binge/purge BN	33 (15.95)		
Atypical EDs	22 (10.6)		
**Co-morbidity (n/p)**			
Depression	39 (18.8)		
ADHD	25 (12.1)		
Anxiety	10 (4.8)		
OCD	9 (4.3)		
PTSD	7 (3.4)		
Dysthymia	4 (1.9)		
Social phobia	3 (1.4)		
Panic disorder	3 (1.4)		
Alcohol abuse	2 (1.0)		
Bipolar disorder	2 (1.0)		
Other	8 (3.9)		

BMI—Body Mass Index; ADHD—Attention Deficit/Hyperactivity Disorder; OCD—Obsessive–Compulsive Disorder; PTSD—Post-Traumatic Stress Disorder.

**Table 2 nutrients-15-04168-t002:** (**a**) Network loadings based on EGA among the clinical group. (**b**) Network loadings based on EGA among the non-clinical group.

(a)					
	Weight Preoccupation	Binge/Purge Behaviors and Concerns of Others	Dieting and Restricting Symptoms	Eating-Related Concerns	
nEAT11	0.35				
nEAT10	0.33				
nEAT1	0.32				
nEAT14	0.30				
nEAT12	0.22				
nEAT3	0.22				
nEAT2	0.18				
nEAT18	0.16				
nEAT8		0.20			
nEAT9		0.18			
nEAT13		0.17			
nEAT4		−0.28			
nEAT16			0.37		
nEAT5			0.31		
nEAT17			0.30		
nEAT15			0.28		
nEAT7			0.26		
nEAT6			0.15		
nEAT23				0.48	
nEAT24				0.44	
nEAT22				0.40	
nEAT21				0.26	
nEAT25				0.25	
nEAT20				0.22	
nEAT26				0.14	
nEAT19				0.09	
(**b**)					
	**Weight Concerns**	**Eating-Related Concerns**	**Food Controls One’s Life**	**One’s Own and Others’** **Control over the Person’s Eating**	**Dieting**
EAT11	0.42				
EAT1	0.29				
EAT14	0.26				
EAT12	0.19				
EAT26		0.33			
EAT24		0.25			
EAT10		0.23			
EAT2		0.17			
EAT9		0.16			
EAT25					
EAT18			0.33		
EAT21			0.32		
EAT4			0.30		
EAT3			0.29		
EAT13				0.33	
EAT8				0.32	
EAT20				0.25	
EAT15				0.22	
EAT5				0.20	
EAT16					0.31
EAT17					0.29
EAT23	0.22				0.25
EAT22					0.23
EAT6					0.15
EAT7					--
EAT19					--

Note. General effect size guidelines for network loadings are 0.15 for small, 0.25 for moderate, and 0.35 for large.

**Table 3 nutrients-15-04168-t003:** Network loadings based on EGA among the non-clinical group (EAT-19).

	Fat Concerns	Eating-Related Concerns	One’s Own and Others’ Control over the Person’s Eating	Dieting
EAT11	0.42			
EAT1	0.30			
EAT14	0.27			
EAT12	0.19			
EAT26		0.37		
EAT24		0.28		
EAT10		0.27		
EAT9		0.18		
EAT2		0.18		
EAT13			0.40	
EAT15			0.27	
EAT8			0.23	
EAT5			0.19	
EAT17				0.32
EAT23	0.25			0.30
EAT16				0.30
EAT22				0.27
EAT7				0.15
EAT6				0.15

Note. General effect size guidelines for network loadings are 0.15 for small, 0.25 for moderate, and 0.35 for large.

**Table 4 nutrients-15-04168-t004:** Network loadings based on EGA among the clinical group (EAT-19).

	Restrictive Weight Concerns	Dieting	Concerns of Others over One’s Eating	Eating-Related Concerns
nEAT11	0.36			
nEAT14	0.35			
nEAT1	0.35			
nEAT10	0.34			
nEAT2	0.20			
nEAT12	0.20			
nEAT17		0.37		
nEAT16		0.37		
nEAT5		0.32		
nEAT15		0.28		
nEAT7		0.26		
nEAT6		0.17		
nEAT9		0.07		
nEAT8			0.24	
nEAT13			0.24	
nEAT23				0.54
nEAT24				0.51
nEAT22				0.40
nEAT26				0.19

Note. General effect size guidelines for network loadings are 0.15 for small, 0.25 for moderate, and 0.35 for large.

## Data Availability

Not applicable.

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
