# Peer review of "Differences in the Factor Structure of the Eating Attitude Test-26 (EAT-26) among Clinical vs. Non-Clinical Adolescent Israeli Females"

_nutrients, 2023, doi:10.3390/nu15194168_

Round 1
Reviewer 1 Report
The manuscript “Differences in the Factor Structure of the Eating Attitude Test-2 26 (EAT-26) among Clinical vs. Non-clinical Israeli Adolescent Females” by Zohar Spivak-Lavi is interesting but there are several changes that must be applied
Please consider about enhancing the outcomes section in the abstract -make them more specific regarding the items that differ
Although interesting, the introduction is very long. This is not a review paper, therefore I would suggest to shorten the introduction with more focus in the EAT26 section and use of the specific questionnaire so far
Section 2.2 is not clear. Please expand text
Table 1 and 2 I would recommend to keep the diagrams with loadings of each item and move tables to appendix. Explanatory text of the tables instead can be kept in the results section
Discussion must be more concise regarding future practical use of the outcomes
Author Response
Reviewer 1
Thanks for the thorough work in reading our paper. Please see below are our references:
Please consider about enhancing the outcomes section in the abstract -make them more specific regarding the items that differ
Thank you. We have addressed this issue. Please see the Results section of the Abstract.
Although interesting, the introduction is very long. This is not a review paper, therefore I would suggest to shorten the introduction with more focus in the EAT26 section and use of the specific questionnaire so far
We have revised the Introduction. Please see p. 2, lines 45-55 & 81-83.
Section 2.2 is not clear. Please expand text.
We have revised this section.
Table 1 and 2 I would recommend to keep the diagrams with loadings of each item and move tables to appendix. Explanatory text of the tables instead can be kept in the results section.
Thank you. We have moved Tables 1 & 2 to the Appendix as suggested. However, we highly recommend keeping things as they were because it is much easier and clearer to understand the details of the graphs with the tables next to them. We leave this decision to you.
Discussion must be more concise regarding future practical use of the outcomes
Thank you. We have revised the Discussion section, please see p. 15.
Reviewer 2 Report
This manuscript describes a study of the validity of the factor structure of the Eating Attitudes Test in clinical and non-clinical adolescent populations. Strengths include that the manuscript is well-written and organized, statistically sound analyses were conducted, and the authors had sufficient sample size for the clinical group.
A few suggestions/comments.
In the introduction, the authors write: “The EAT stands out as the most accepted tool in defining disordered eating attitude 77 symptoms in the general population” is an opinion of the authors rather than an accepted fact. While the EAT is a frequently used and well-validated measure, this statement seems inappropriate.
-The reasoning for Hypothesis 3 is unclear and not supported by the introduction. Why do the authors suspect that 20 will no longer be a valid cutoff?
-Typically, the suggested sample size for CFA is 200 as results can be less stable with smaller sample sizes. Though sample size was adequate for the clinical group, are the authors able to add to the non-clinical sample size? If not, this should be noted as a significant limitation and results for the non-clinical sample should be couches as preliminary at best.
Some repetition in the Discussion, e.g., information about inconsistent findings across ethnic groups in Israel appears nearly verbatim in Introduction and Discussion
Author Response
Thanks for the thorough work in reading our paper. Please see below are our references.
Reviewer 2
This manuscript describes a study of the validity of the factor structure of the Eating Attitudes Test in clinical and non-clinical adolescent populations. Strengths include that the manuscript is well-written and organized, statistically sound analyses were conducted, and the authors had sufficient sample size for the clinical group.
A few suggestions/comments.
In the introduction, the authors write: “The EAT stands out as the most accepted tool in defining disordered eating attitude 77 symptoms in the general population” is an opinion of the authors rather than an accepted fact. While the EAT is a frequently used and well-validated measure, this statement seems inappropriate.
Thank you for your observation. We have changed the wording and believe you will find it more appropriate now.
-The reasoning for Hypothesis 3 is unclear and not supported by the introduction. Why do the authors suspect that 20 will no longer be a valid cutoff?
Thank you. We clarified Hypothesis 3 in accordance with your comment, please see p. 4. Additionally, we clarified and emphasized the rationale for this hypothesis, please see p. 3.
-Typically, the suggested sample size for CFA is 200 as results can be less stable with smaller sample sizes. Though sample size was adequate for the clinical group, are the authors able to add to the non-clinical sample size? If not, this should be noted as a significant limitation and results for the non-clinical sample should be couches as preliminary at best.
Thank you for this important comment. As suggested, we have added this point to the Limitations section.
Some repetition in the Discussion, e.g., information about inconsistent findings across ethnic groups in Israel appears nearly verbatim in Introduction and Discussion
Thank you. We have omitted the repetition from the Discussion. Please see p.14, line 425-426